# Ta_2_O_5_ Nanocrystals Strengthened Mechanical, Magnetic, and Radiation Shielding Properties of Heavy Metal Oxide Glass

**DOI:** 10.3390/molecules26154494

**Published:** 2021-07-26

**Authors:** Xinhai Zhang, Qiuling Chen, Shouhua Zhang

**Affiliations:** 1School of Traffic and Materials Engineering, Hebi Polytechnic, Hebi 458030, China; doya925@gmail.com (X.Z.); disney.us@gmail.com (S.Z.); 2Material Sciences & Engineering Department, Henan University of Technology, Lianhua Road 100, Zhengzhou 450001, China

**Keywords:** Ta_2_O_5_, mechanical property, radiation shielding, heavy metal oxide glass

## Abstract

In this study, for the first time, diamagnetic *5d*^0^ Ta^5+^ ions and Ta_2_O_5_ nanocrystals were utilized to enhance the structural, mechanical, magnetic, and radiation shielding of heavy metal oxide glasses. Transparent Ta_2_O_5_ nanocrystal-doped heavy metal oxide glasses were obtained, and the embedded Ta_2_O_5_ nanocrystals had sizes ranging from 20 to 30 nm. The structural analysis of the Ta_2_O_5_ nanocrystal displays the transformation from hexagonal to orthorhombic Ta_2_O_5_. Structures of doped glasses were studied through X-ray diffraction and infrared and Raman spectra, which reveal that Ta_2_O_5_ exists in highly doped glass as TaO_6_ octahedral units, acting as a network modifier. Ta^5+^ ions strengthened the network connectivity of 1–5% Ta_2_O_5_-doped glasses, but Ta^5+^ acted as a network modifier in a 10% doped sample and changed the frame coordination units of the glass. All Ta_2_O_5_-doped glasses exhibited improved Vicker’s hardness, magnetization (9.53 × 10^−6^ emu/mol), and radiation shielding behaviors (*RPE*% = 96–98.8%, *MAC* = 32.012 cm^2^/g, *MFP* = 5.02 cm, *HVL* = 0.0035–3.322 cm, and *Z_eff_* = 30.5) due to the increase in density and polarizability of the Ta_2_O_5_ nanocrystals.

## 1. Introduction

Transition metal oxides have been widely incorporated in vitreous materials due to optical changes and property enhancements to vitreous networks. In addition, transition metal oxides also strongly increase the chemical and thermal stabilities of glass formers because of their intermediary behavior related to a high coordination number of the metallic ions, resulting in higher connectivity of the glass network [1]. Among transition metal oxides, Tantalum oxide (Ta_2_O_5_) is an important tunable band gap and high-κ (>20) dielectric material with interesting structural and functional properties [2]. Ta^5+^ ions, with an empty *d* shell (*d*^0^ ions) having completely occupied the outermost electronic shell, can contribute to the diamagnetic character [3]. In addition, Ta_2_O_5_ has a high density, large refractive index, and a temperature-dependent structure; these features allow it to have potential applications in mechanical stability [4,5] and radiation shielding [6] applications. 

However, except for several Ta_2_O_5_ thin films, Ta_2_O_5_ has been less investigated due to its lower solubility in classical glass formers, as well as its high melting temperature (1825 °C). The studies of Ta_2_O_5_ are limited to several Ta_2_O_5_-doped silicate [7], phosphate [8,9,10], germinate [11], and tellurite [12] glasses for coating, photovoltaic, crystallization, and dielectric studies. Rare Ta_2_O_5_-doped heavy metal oxide glasses are reported, except recently published Bi_2_O_3_–TeO_2_–ZnO glasses [12] and ZnO–Ta_2_O_5_–TeO_2_ [8] by the Gokhan Kilic group. 

Usually, transition metal ions act as glass network modifiers in glasses and provide a higher coordination number [13]. However, Ta_2_O_5_ is known as a conditional and poor glass “network-former’’, compared with silica and phosphorus pentoxide. Ta_2_O_5_ is an intriguing material because it is a “modifier” oxide composed of high field-strength cations, which has not been melt-quenched to form glass but is typically formed by ion beam sputtering [8]. At a lower content, it acts as a glass former, which increases the network connectivity with higher rigidity, resulting in a higher glass transition temperature and larger thermal stability against devitrification in phosphate glass [9] and 93GeO_2_–6Ta_2_O_5_–Bi_2_O_3_ glass [11]. Ta_2_O_5_ has been found to be a modifier, in most cases, at a higher content, which exists as TaO_6_ units [13,14]. In this case, the empty or unfilled d-orbital (outer electronic configuration *5d*^0^*6s*^0^ of Ta^5+^ ions strongly contribute to the large ionic refraction (23.4) and large refractive index and polarizability [15]. These features are attractive for magnetic properties. 

Recently, the radiation shielding property of telluride glasses [16,17,18,19,20] and polymer [21] has generated much interesting research due to the introduction of highly dense and polarizable oxides, such as Ag_2_O, TeO_2_, Sb_2_O_3_, and so forth. Ta_2_O_5_ is a high density (8.2 g/cm³) material; the high density of Ta_2_O_5_-doped glass originated from the large packing effect of Ta in glass matrix [22] since the energy loss per traveled distance of gamma rays and charged ionizing particles (electrons, protons, alpha particles, ions) is proportional to electron density in the matter. Therefore, Ta_2_O_5_-doped transparent glasses with high densities are attractive in radiation shielding and a protective application in nuclear energy production and medicine [23,24]. Primarily, gamma radiation and X-rays emitted from the nuclear reactors with an elevated frequency are lethal since the interaction of ionizing radiation with materials requires dense elements with high atomic numbers, lower relaxation lengths, and lower half-value layers to be good radiation shielding materials [23]. Thus, glass components that have heavy metal oxides (e.g., PbO, Bi_2_O_3_) show excellent shielding properties under gamma and X-ray [25]. Two Ta_2_O_5_-doped tellurides [8,12] glasses and one Ta_2_O_5_-doped borate glass [22] were reported recently for radiation shielding study. However, Ta_2_O_5_-doped heavy metal oxide glass have still not been investigated, but it is expected to be the most ideal candidate for shielding applications. 

The influence of the high polarizability and high refractive index (n = 2.18 at 550 nm) of Ta_2_O_5_ is not clear in terms of the radiation shielding of glass. Similar to Ta_2_O_5_, PbO plays dual roles in glass-forming as well. The network former Pb^2+^ ions impart a three-dimensional spatial network character to the glass. In fact, the easily polarizable valence shell of the Pb^2+^ ion strongly interacts with the highly polarizable O^2-^ ion, giving rise to a rather covalent Pb–O bond [26]. The highly polarizable ions (Bi^3+^), from modifier Bi_2_O_3_ to glass, can prevent melt crystallization by the asymmetry structure resulting from oxygen polyhedra. Bi_2_O_3_ is found to exist as BiO_3_ pyramidal and BiO_6_ octahedral units [27]. The bismuth borate system has specific properties, such as non-hygroscopic, high density, high optical basicity, large polarizability, and a high refractive index. In addition, the phonon energy of heavy metal oxide glass can enhance the chemical durability and radiation shielding performance [28]. 

On the other hand, nanocrystal-containing glasses have attracted research attention in recent years because nanocrystals have a large surface area and, therefore, they are much more reactive than their bulk counterparts [29]. The doping of nanocrystals inside glass can greatly improve the mechanical performance of glasses [30], which is good for future device fabrication. In this study, Ta_2_O_5_ nanocrystals were synthesized using the hydrothermal method, and their influence on glass structure, specifically, their mechanical, magnetic, and radiation shielding of Ta_2_O_5_-doped heavy metal oxide diamagnetic glasses, was investigated. 

## 2. Experiments

### 2.1. Synthesis of Ta_2_O_5_ Nanocrystal

In a typical synthesis procedure, 0.4888 g of tantalum pentachloride TaCl_5_ (99.8% Sigma Aldrich) was added to 45 mL of anhydrous benzyl alcohol. The solution was firstly subjected to ultrasound for 20 min and then magnetically stirred for 30 min to get a completely dissolved mixture. The mixture was then transferred to a 100 mL Teflon-lined stainless steel autoclave and carefully sealed. The solvothermal reaction in the autoclave was heated to 220 °C by 2 °C/min and kept for 72 h. After cooling to room temperature, the resulted cloudy suspensions were centrifuged at 4500 rpm for 15 min to retrieve the product. Repeated washing with acetone and ethyl alcohol was performed to remove possible organic impurities. The product was subsequently dried in air at 70 °C for 4 h. The off-white final product was thermally treated at 600, 800, and 1000 °C by 10 °C/min and residual for 5 h.

### 2.2. Fabrication of Ta_2_O_5_-Doped Glasses

Transparent glasses were fabricated by using reagent grade PbO, Bi_2_O_3_, H_3_BO_3_, and as-synthesized Ta_2_O_5_, which was prepared at 600 °C with a size of around 30 nm. The stoichiometric compositions of the batch materials (20 g) were 40PbO–45Bi_2_O_3_–(15-x) H_3_BO_3_–xTa_2_O_5_, where x = 1, 5, and 10 mol.%; the corresponding glasses were labeled as PBT_0_ (host), PBT_1_, PBT_5_, and PBT_10_. The batch was well mixed in a mortar and the mixture was melted in a gold (95% Pt–5% Au) crucible at 1000 °C for one hour. The glasses were obtained by quenching the melt on a brass mold, and then subjected to annealing at 50 °C higher than the glass transition temperature for 4 h to release the thermal stress that was produced during the quenching process. 

### 2.3. Characterization of Samples 

The crystallinity and phase of samples were characterized by X-ray diffraction (Bruker, D8 Discover system) with a Cu-Kα 1.54056 Å wavelength X-ray source. The morphologies and sizes of the nanocrystals were examined by scanning electron microscopy (TEM, FEI, Quantainspect 200). Raman and Fourier-transform infrared spectroscopy (FT-IR) spectra were recorded using an MKI Renishaw Raman spectroscopy and Varian Cary500, respectively, with a resolution range of ±0.5 cm^−1^ and a KBr medium prepared with 0.5 wt% of the sample. The Vickers hardness of the glasses was tested using a 136° pyramidal diamond indenter applied to the glass at a weight load of 150 g. According to the square indent formed on the glass surface, Vickers hardness can be calculated through the following formula: HV = 0.1891*F*/*d*^2^, where *F* is the applied load and *d* is the diagonal of the indentation. The chemical valence energy was recorded using X-ray photoelectron scanning (XPS, ESCALAB 250) spectroscopy with a monochromatic Al K X-ray source (*hv* = 1486.6 eV). The obtained binding energies (BEs) were calibrated with that of an adventitious carbon (C1s) core level peak at 284.6 eV as a reference. The magnetic property of the glasses was evaluated by a vibrating sample magnetometer (VSM) instrument. 

A radiation shielding characteristics test was carried out to assess the gamma ray shielding capacity of the glasses [28]. A 10 cm collimator of lead and silt with a 2 cm diameter was used to collimate the incident gamma photons emitted from a mono-energetic gamma ray source of 10^6^ photons per minute. Figure 1 shows the schematic diagram of the radiation shielding measurement. The distance between the detector and the source was 15 cm. The mass attenuation coefficient (*MAC*) of lead used in this instrument is 1.306 × 10^2^ (0.015 MeV), 7.102 × 10^−2^ (1 MeV), and 4.972 × 10^−2^ (10 MeV). The half-value layer (*HVL*) of lead is 4.8 mm and the *Z_eff_* of lead is 82.

The Monte Carlo (MCNP-5) simulation code was applied to assess the gamma ray shielding capacity for the fabricated glasses [14]. The MNCP simulation geometry and the input file are shown in Figure 1. The applied geometry was shielded from the outer environment by a cylinder of lead with a 5 cm thickness. Then, a collimator of lead with a length of 10 cm and a silt diameter of 2 cm was used to collimate the incident gamma photons emitted from a mono-energetic gamma ray source. The mono-energetic source was set up to emit 10^6^ photons per min in the +Z direction. The made glasses were placed between the gamma ray supply and the detector. In the present simulation process, glass in this study acted as a shielding material, while bulk Pb blocks avoided the scattered photons. The F4 tally mesh detector was used to count the high-energy photons intensities and estimate the average track length. The glass was located between the radiation source and the detector, and the distance between the source and the detector was 15 cm. The simulation was carried out, and the recorded data has a relative error of less than 1%, as shown in the output sheets of the MCNP code [15]. 

## 3. Results and Discussion

### 3.1. Structure of Ta_2_O_5_ Nanocrystals

The physical properties of nanocrystal-doped glasses depend on the size, shape, and properties of the nanocrystals. Therefore, we firstly investigated the structure of synthesized nanocrystal. Figure 2a–c show the XRD pattern of samples prepared at 600 °C, 800 °C, and 1000 °C, respectively. Both the 600 °C and 800 °C obtained samples show intense diffraction peaks at 2θ = 22.75°, 24.88 °, 36.79°, 46.78°, 50.35°, 55.70°, 58.90°, and 64.03°; these peaks correspond well to the planes of (001), (100), (101), (002), (110), (102), (200), and (201) of the hexagonal phase of β-Ta_2_O_5_, respectively (JCPDS # 0019-1299) [31]. The lattice parameters a = b = 3.6239Ả, c = 3.8803, α = β = 90°, and γ = 120° belong to the spatial group P6/mmm [28]. This XRD pattern indicates that Ta_2_O_5_ prepared at 600 °C and 800 °C is well-crystallized as hexagonal δ-Ta_2_O_5_ phase [30]. The sample produced at 1000 °C shows diffraction peaks at 2θ = 22.84°, 28.40°, 36.74°, 44.75°, 46.75°, 49.75°, 50.70°, 55.68°, 58.44°, and 63.87°, relating to the (001), (1110), (1111), (340), (002), (0220), (2151), (1112), (2220), and (2221) planes of an orthorhombic phase of β-Ta_2_O_5_, respectively (JCPDS # 25-0922).

The lattice parameters a = 6.1982Ả, b = 3.6629Ả, c = 3.8900, and α = β=γ = 90° belong to the spatial group A/mm^2^ [8]. The phase change from hexagonal into orthorhombic is evidenced by the diffraction peak splitting at 2θ = 28.38° (100) into 2θ = 28.22° (1100) and 28.40° (1110). Another splitting of the peak occurs at 2θ = 50.35° (110), which is split into 50.70° (2151) and 49.75° (0220). The (340) plane of orthorhombic β-Ta_2_O_5_ at 2θ = 44.75° is very weak in this study.

The average size of the nanocrystals was calculated using the Debye–Scherrer Equation (1):
(1)$$D=\frac{K\lambda }{\beta cos\theta }$$
where *K* is a dimensionless shape factor of 0.9, *λ* is the X-ray wavelength, *β* is the line broadening at half the maximum intensity, and *θ* is the Bragg angle. After calculation, the sizes of the nanocrystals obtained at 600 °C, 800 °C, and 1000 °C temperatures were 20.8, 21.06, and 22.12 nm, respectively. The higher temperatures accelerated the aggregation of nanocrystals and induced an increase in size. On the other hand, the highly polarized Ta^5+^ ions also contributed to the increase in the nanocrystal size. 

Raman spectroscopy is commonly used in chemistry to provide a structural fingerprint by which molecules can be identified. In order to give an in-depth study on the structure of Ta_2_O_5_ nanocrystals, Raman spectra were recorded, which provided characteristic vibration modes of Ta_2_O_5_ nanocrystals. Figure 2d–f show the Raman spectra of Ta_2_O_5_ prepared at 600 °C, 800 °C, and 1000 °C, respectively, which show the structural phase transition from low-symmetry hexagonal to orthorhombic Ta_2_O_5_. Generally, the low-energy < 100 cm^−1^ phonon modes originate from interactions between different Ta polyhedral and Ta_6_O_12_^6+^ clusters [31]. The mid-energy Raman bands (100–400 cm^−1^) correspond to O–Ta–O bending vibrations in octahedral TaO_5_. The higher energy bands (400–800 cm^−1^) could be associated with the coupled modes mainly involving the stretching of various Ta–O bonds. The higher wavenumber > 800 cm^−1^ is related to the stretching vibration of Ta–O–Ta bonds [32]. It can be seen from Figure 2f that the lowest wavenumber vibration Bands 1 to 3 are sharp and intense in Ta_2_O_5_ prepared at 1000 °C, while these bands disappear for Ta_2_O_5_ prepared at 600 °C and 800 °C. These results indicate that the Ta polyhedral and Ta_6_O_12_^6+^ clusters existed in Ta_2_O_5_ prepared at 1000 °C due to the higher temperature. The vibration information are provided in Table 1.

Therefore, from the comparison of the XRD and Raman spectra of three samples, the Ta_2_O_5_ prepared at 800 °C exhibits promising size, morphology, and structure, and this sample was used as a dopant for the glasses.

### 3.2. Morphology and Structure of Ta_2_O_3_-Doped Glasses

The morphology and nanocrystal distribution inside glass are important for glass properties. Therefore, TEM images of glasses were taken to provide the morphology and distribution of the nanocrystals. As mentioned in the experimental part, the Ta_2_O_5_ prepared at 600 °C with a size of about 20 nm was selected as the dopant to glasses, only varying with the dopant amounts of 1%, 5%, and 10%. After doping into the glass, different amounts of Ta_2_O_5_ experienced the 1000 °C melting, combined with the deformation of space and pressure resulting from the quick temperature gradient. Therefore, doped nanocrystals (same size before doping) presented differences in size, shape, solubility, and even crystal phase. Transparent and yellowish glasses containing Ta_2_O_5_ nanocrystals were obtained and the photographs and SEM morphology images are shown in Figure 3a–c. All glasses contain nanocrystals with sizes ranging from 20 nm (PBT_1_) to 24 nm (PBT_5_ and PBT_10_). The presence of nanocrystals and the varying crystal sizes result in the transparency slightly decreasing as the doping amount increases. The insets of Figure 3a–c shows the nanocrystal distribution inside the glass matrix, which confirms the particle sizes from the SEM observations, revealing that the nanocrystal size is not doping concentration-dependent.

The structure of glass after the doping of the Ta_2_O_5_ nanocrystals inside the glasses was influential to the properties. The structures of the glasses were studied using XRD, FT-IR, and Raman spectra, which are shown in Figure 4a–d, respectively. It can be seen that glass PBT_1_ shows a halo and a broad main peak in Figure 4a at 28.84°, illustrating the amorphous glassy nature. However, small and weak signals of peaks appear at 2θ = 22.84°, 28.40°, 36.74°, 46.75°, 50.70°, 55.68°, and 58.44°. According to the study on JCPDS #25-0922, they are related to the (001), (1110), (1111), (002), (0220), (0221), and (2220) planes of an orthorhombic phase of β-Ta_2_O_5_, respectively [33]. These peaks also appear in PBT_5_ with a slightly increasing intensity. These observations suggest that even though undissolved Ta_2_O_5_ crystals existed, the whole matrix of PBT_5_ still has a glassy character. Different from PBT_1_ and PBT_5_, PBT_10_ exhibits sharp and intense XRD peaks, suggesting an obvious crystalline tendency in the matrix, which was probably caused by the highest undissolved Ta_2_O_5_ nanocrystals in the matrix.

Figure 4b shows the FT-IR spectra of three samples, which show 7 absorption peaks (and 2 more in the inset). The absorption peak at 422 cm^−1^ was due to the bending vibration of the tetrahedral PbO_4_ groups, and peaks at 473 cm^−1^ and 719 cm^−1^ are attributed to the bending and symmetric stretching vibration of Bi–O bonds in BiO_3_ pyramidal units, respectively [34]. The weak peak at 908 cm^−1^ was caused by the stretching vibration of B–O bonds in BO_4_ units, which is apparent in PBT_1_ but gradually weakens as the doping amount increased [35]. The peak at 1160 cm^−1^ is from the asymmetric stretching vibration of B–O bonds in BO_3_ pyramidal units, which shifted to a lower wavenumber side as the doping amount increased, indicating the conversion of BO_4_ to BO_3_ [35]. Considering that all glasses were fabricated in air, water can be present in glasses. Other peaks at 2911 (very weak) and 3463 cm^−1^ are related to the stretching vibration of OH bonds in the samples. A weak peak around 1630 cm^−1^ is due to the bending vibration mode of OH groups [36]. The inset of Figure 6b shows the amplified wavenumber between 600–700 cm^−1^; one peak at 668 cm^−1^ is related to the Ta–O–Ta vibration in TaO_6_ octahedral units, which appears in three samples, but peak at 648 cm^−1^, related to the tantalum clusters, appears only in PBT_10_, which usually appears at a high Ta_2_O_5_-concentrated sample [9]. The appearance of a peak at 648 cm^−1^ also indicates that TaO_6_ aggregates in a network, which could rapidly increase the size and numbers of Ta_2_O_5_ nanocrystals. This explains why the grain size and numbers of Ta_2_O_5_ in PBT_10_ are obviously larger than others in the TEM observation (Figure 3).

Figure 4c shows the Raman spectra of three samples between wavenumbers of 0 and 700 cm^−1^. 7 Raman bands with different intensities at 73, 99, 124, 250, 314, 537, and 660 cm^−1^ were observed in amplified Raman spectra (Figure 4d). The band at 73 cm^−1^ and 99 cm^−1^ belong to the boson peaks of glass, while the band at 124 cm^−1^ is attributed to the heavy metal Pb–O symmetric stretching vibration in PbO_4_ tetrahedral groups [37]. The band at 314 cm^−1^ is due to the Bi–O–Pb, Pb–O–Pb, Pb–O–Bi stretching linkages. The band centered at 537 cm^−1^ is attributed to the stretching vibration of Bi–O in BiO_3_ pyramidal units. The Raman frequency around 250 cm^−1^ is assigned to O–Ta–O bending vibration in TaO_6_ octahedral units, while the band at 660 cm^−1^ is from the stretching of Ta–O–Ta linkage vibrations [32], and the intensities of both bands increased with the doping amount, indicating that Ta_2_O_5_ nanocrystals exist as TaO_6_ units in a glass network.

The appearance of Ta–O–Ta vibrations and corresponding changes to BO_4_ and BO_3_ suggest that the tantalum atoms in PBT_1_ participated in the glass network, resulting in higher network connectivity due to the high BO_4_ content. PBT_5_ remains glassy in nature even with more Ta_2_O_5_ crystals. A large amount of Ta atoms in PBT_10_ bonded to the TaO_6_ units are responsible for forming tantalum oxide-rich regions (tantalum clusters), promoting the deformation and distortion of the network.

### 3.3. Chemical and Physical Properties

In order to verify the roles of tantalum in the structures of different samples in terms of the oxygen bonds of the network, XPS analysis was carried out and the results are shown in Figure 5. Figure 5a is the outline of the XPS core level energy of the Pb4f, Pb4d, Bi4f, Bi4d, Ta4d, Ta4p, O1s, and B1s of the PBT_10_ sample. Among them, the O1s and B1s are most sensitive to structural changes induced by the doping of Ta_2_O_5_. It is well known that tetrahedral BO_4_ units play an important role in strengthening the network, while on the contrary, triangular BO_3_ units are not helpful for glass stability [38]. For the same reason, glass with good connectivity would have more bridging oxygen bonds (BO). When these BOs are broken into non-bridging oxygen bonds (NBO) by modifiers or other impurities, glasses will lose their stability and present poor chemical, mechanical, and thermal properties. In this context, the O1s and B1s were studied to get the coordination and oxygen bonds information.

Figure 5b shows the B1s spectrum with the main peak located at the binding energy range of 190–194 eV. With the increase of Ta_2_O_5_ amounts_,_ the whole peak progressively shifts toward the lower binding energy side (gray dotted line for a guide). For example, the binding energy of B1s of PBT_1_ is located at 192.7 eV, while it shifts to 192.3 eV for PBT_5_ and 191.9 eV in PBT_10_. It is well known that B^3+^ ions can form two coordination bonds (BO_3_ and BO_4_) in a glass network, and the BO_4_ units have higher binding energy than that of BO_3_ groups [39]. Therefore, the decrease in binding energy indicates a decrease in BO_4_ and an increase in BO_3_. This conclusion is also confirmed by the deconvolution of B1s peaks. It can be seen that the area of BO_4_ gradually decreased with the doping amount and reached the minimum in PBT_10_; meanwhile, the BO_3_ increased. This result matches well with the observation of the FT-IR bands at 908 cm^−1^. Since triangular BO_3_ usually is less stable than tetrahedral BO_4_, PBT_1_ and PBT_5_ glasses are more stable than PBT_10_.

The conversion of BO_4_ to BO_3_ coordination numbers will yield free non-bridging oxygen and, accordingly, influences the oxygen bonds in glass. Figure 5c shows the core level energy of O1s with Gaussian deconvolutions. As shown in Figure 5c, there is one main peak located at the 527–535 eV binding energy of the O1s spectra in PBT_1_ and PBT_5_ glass samples. According to previous works [26], the main peak of the O1s spectrum has two contributions: bridging oxygen (BO) and NBO atoms. The NBO atoms are located at low binding energies of 529–531 eV, while BO atoms correspond to higher binding energies of 531–533 eV [26]. From Figure 4c, the O1s XPS spectra remarkably shift towards lower energies, demonstrating that excessive Ta_2_O_5_ increases the degree of polymerization of glass structure.

As can be seen from TEM images in Figure 3, even though all samples contain nanocrystals, the structure deformation of PBT_10_ is the most remarkable, which provides low viscosity and quicker heat flow through the matrix, and speeds up the growth of residual Ta_2_O crystals, leading to the rapid increase in size and/or numbers. These residual nanocrystals are regarded as nucleating centers in the matrix and, therefore, they decrease the ability against devitrification.

Figure 6 shows the relationship between Vickers hardness of the non-bridging oxygen numbers and other physical parameters. Structural properties, such as the density, oxygen packing density, and optical basicity values, were calculated using the expressions reported in the literature [9] and interpreted with the physical properties of Ta_2_O_5_-embedded glasses. From Table 2 and Figure 6, apparently, the present samples’ density and optical basicity increased progressively with the addition of Ta_2_O_5_ content, indicating the high compactness of the glass structure. The higher molecular weight and higher optical basicity of Ta_2_O_5_ helped to enhance these two parameters. The oxygen packing density (*OPD*) firstly increased and then decreased with the increase in the doping amount, indicating that the non-bridging oxygen occurred for TPB_10_ glass. This matches well with the XPS observation. From the comparison of physical parameters, the TPB_5_ (green dash circle in Figure 6b) presents promising physical and mechanical performance.

Figure 7 shows the Vickers hardness test indentations on the glasses. It can be seen that the host glass, without any crystals, presents a complete indentation; its edges and surrounding surface are not damaged and the Vickers hardness is 467.98 HV. Similar indentation profiles can be seen for the PBT_1_ and PBT_5_ glasses with an increase of hardness, respectively. Such enhancement of the hardness for the TPB_1_ and TPB_5_ glasses is mainly due to the network former role of Ta_2_O_5_, as confirmed by the Raman and FT-IR studies, which strengthened the crosslink of the network through its big molecule weight and network former nature. However, the PBT_10_ presents a much bigger indentation with the surrounding surface seriously damaged, and the hardness decreased to 446.88 HV as well. The deterioration of mechanical properties of PBT_10_ was caused by the large nanocrystals and tantalum clusters (refer to the Raman analysis) inside the matrix, which cracked the nucleus and broke the homogeneity of the matrix. When a load is applied, cracks spread to its surrounding area. This also proves that the doping amount of nanocrystals higher than 5% is not good for glass mechanical stability.

### 3.4. Radiation Shielding Property

From the above studies on density, *OPD*, and hardness, it is found that that Ta_2_O_5_ nanocrystals enhanced the compactness and mechanical performance, which are helpful for improving the radiation shielding as well. Therefore, the radiation shielding properties were studied, among which the attenuation coefficient is important. The linear attenuation coefficient values (*µ*) indicate the probability of eliminating a photon that occurred due to the exposure of the sample to certain energy per path unit. In this study, the *µ* was measured experimentally at 0.015, 0.05, 0.08, 0.3, 0.5, 3, 5, and 10 MeV photon energies by using the Beer–Lambert Equation [26]:(2)$$\mu =\frac{\ln\left(\frac{I_{0}}{I}\right)}{t}\;{\mathrm{cm}}^{-1}$$
where *I* is the intensity of the transmitted *γ* ray and *t* is the sample thickness. *µ_m_* is related to *µ* and glass density ρ according to Equation (3) [39]:(3)$$\mu_{m}=\frac{\mu }{\rho }=\sum_{i}f_{i}{\left(\frac{\mu }{\rho }\right)}_{i}\;{\mathrm{cm}}^{2}/g$$
where *f_i_* is the weight fraction and ($\frac{\mu }{\rho })i$ is the mass attenuation coefficient *MAC* of the *i*th element [28]. The relative deviation (*RD*) between the data and simulation can be calculated from Equation (4). The calculated data are reported in Table 3.
(4)$$RD=\frac{\left({\left(\mu /\rho \right)}_{MCNP}-{\left(\mu /\rho \right)}_{XCOM}\right)}{{\left(\mu /\rho \right)}_{MCNP}}\times 100\%$$

Figure 8a plots the *MAC* versus different radiation energy points, which reached the maximum level at 0.015 MeV, varying in the range of 26.854–32.012 cm^2^/g for PBT_1_ to PBT_10_, respectively. Gamma rays interacted with the matter by photoelectric absorption (PE), Compton scattering (CS), and pair production [18,19,20,23] which contributed to the *µ_m_* value. Due to the prevalence of photoelectric interaction in the low energies, the values of *µ_m_* suffered a rapid reduction with the increase of photon energies. This fast drop trend was caused by the PE cross-section varied with E^−3.5^ [16,17,18,19,20,40]. At around 0.08 MeV, the *MAC* values for all glasses had abrupt progress due to the X-ray K-edges of Pb and Bi, and the highest absorption peak kept constant due to the fixed molar content of Pb^2+^ and Bi^3+^ in all the samples. At higher gamma photon energy (0.3–3 MeV), the probabilities of the CS interaction increased and predominates. Thus, the *µ_m_* values have a moderate drop trend with an increase in the incident gamma photon energy because the CS cross-section varied with E^−1^, which can be seen from Equation (5).

The Compton Scattering effect can be expressed by the following formula:(5)$$\frac{1}{E^{\prime }}-\frac{1}{E}=\frac{1}{E_{0}}\left(1-\cos\theta \right)$$
where *E*′ is the energy of the scattered photon, *E* is the energy of the incident photon, and *θ* is the scattering angle. The *E* and *E*′ as a function of *θ* can be easily measured with a photomultiplier detector and multichannel analyzer system. A plot of measurements of $\frac{1}{E^{\prime }}-\frac{1}{E}$ versus measurements (1−cosθ) should result in an almost linear graph slope as the inverse of the electron’s rest energy $\frac{1}{E_{0}}$.

Figure 8b shows the energy-dependent cross-section of the glasses. It is known that the interaction cross-sections for the pair production vary with log I [41]. The minimum µ_m_ value appears at higher gamma photon energies of 10 MeV, varying between 0.026 and 0.028 cm^2^/g. From Table 3, the *µ_m_* simulated by MCNP matches well with the XCOM database with an *RD* lower than 10%. Therefore, the radiation shielding effect of Ta_2_O_5_ doping is much more remarkable at low photon energies, and the CS process domination eliminated the effect at high photon energies. In addition, at each photon energy point, the *µ_m_* value of the sample increased with the doping amount, indicating that the Ta_2_O_5_ doping enhanced the shielding ability.

The mean free path (MFP), defined as *MFP* = 1*/µ*, was evaluated regarding the gamma photons’ simulated average track length for Ta_2_O_5_-doped glasses. Figure 9 displays the MFP at different incident gamma photon energies. We noticed that the lowest MFP values appeared at 0.015 MeV gamma photon energy with 0.008 and 0.010 cm MFP for all glasses. After that, owing to the PE interaction in low gamma photon energy, the MFP values increased rapidly with the increase of gamma photon energy. Due to the predomination of the CS interaction over 0.3 MeV, the MFP increased moderately with the rise of the gamma photon between 0.3 to 1 MeV. When the energy reached several MeV, the MFP varied slowly with increasing of energy. Subsequently, the MFP began to decrease when the energy was above 10 MeV due to the PC interaction. The MFP’s simulated values were augmented to the higher MFP around 10 MeV and were between 4.14, 5.02, and 5.58 cm for the TPB_1_, TPB_5_, and TPB_10_ glasses, respectively, indicating that the doping of Ta_2_O_5_ decreases the MFP value.

To understand the attenuation behavior of material composed of different elements, it is necessary to study its effective atomic number (*Z_eff_*), which is directly related to the interaction of matter with radiation. The *Z_eff_* of glass can be calculated by the following formula [22]:(6)$$Z_{eff}=\frac{\sum_{i}f_{i}A_{i}{\left(\frac{\mu }{\rho }\right)}_{i}}{\sum_{j}\frac{A_{j}}{Z_{j}}{\left(\frac{\mu }{\rho }\right)}_{j}}$$
where *Z_j_, f_j_*, and *A_j_* denote an atomic number of the *i*th constituent element, the factional percentage, and the atomic weight, respectively. The trend of *Z_eff_* at different energy ranging from 0.015–10 MeV is plotted in Figure 10a. For gamma photon energy, in which the PE interaction is essential, *Z_eff_*’s values were observed to decrease speedily with a rise in the incident gamma ray energy. This speedily decreased because the PE cross-section was inversely proportional to E^3.5^ [28]. Above 0.8 MeV, where the CS was the primary interaction, *Z_eff_* reduced moderately with an increase in the gamma photon energy. Above several MeV, where the PC was the primary interaction, *Z_eff_* began to slowly increase, with a boost in gamma photon energy due to the PC cross-section, which varied with log^E^ [24]. The maximum *Z_eff_* values were obtained around 0.089 MeV due to the K-edges of the lead and bismuth elements. Among the present samples, the PBT_10_ sample exhibits the highest *Z_eff_* due to the highest Ta_2_O_5_ content, indicating that Ta_2_O_5_ increases the interaction between glass and radiation energy, resulting in less energy leaking out to the environment.

The half-value layer (*HVL*) is the thickness of the materials that rescues the intensity of radiation to half its original value, which is defined as *HVL = ln*2*/µ*. A lower *HVL* value points out high gamma rays’ shielding capacity. Figure 10b shows the plot of the *HVLs* of the glasses at different energy points, which prominently decreased as the Ta_2_O_5_ doping amount increased. On the other hand, the *HVL* value increased obviously with the energy of gamma photons.

From above studies, it can be seen that the highest *MAC*, *HVL*, and *Z_eff_* of the present study are 32.012 cm^2^/g, 0.0035–3.322 cm, and 30.5 at 0.015 eV, respectively. These data are reasonable and good if we compare them to the pure lead bulk (*MAC*, *HVL*, and *Z_eff_* of 130.6 cm^2^/g, 4.8 mm, and 82) at the same at 0.015 eV energy.

The radiation protection efficiency (*PRE*) of the samples was calculated with the base *µ* and glass thickness (*t*), according to Equation (7):(7)$$PRE=\left(1-e^{-\mu t}\right)\times 100$$

Figure 10c shows the plot of the calculated *RPE*% at four fixed gamma ray energies (0.015, 0.5, 5, and 10 MeV) with a thickness of 1 cm. It is clear that for low gamma ray energy (0.015 MeV) where the PE was the main interaction, values of the *RPE* reached the maximum and varied between 96 and 98.8% for the three samples, indicating shielding to most of the incoming photons. With an increase in the incident gamma photon energy, the *RPE%* gradually decreased for the fabricated glasses until 12.5–18.9% for 10 MeV radiation. From Figure 10c, the substitution of B_2_O_3_ with Ta_2_O_5_ significantly improved the *RPE%* of heavy metal oxide glasses (green curves in Figure 10c) due to the high density and big mass of Ta_2_O_5_.

### 3.5. Magnetic Property

Usually, materials doped with a large mass dopant such as Ta_2_O_5_ exhibit an improved polarization character, which is helpful for magnetic and magneto-optical performance. On the other hand, glass with good magnetic and large compactness properties has generated much interest in recent years for radiation shielding and magneto-optical devices. In the case of this study, Ta^5+^ has an empty *d* shell (*d*^0^ ions) and a completely occupied outermost electronic shell which are appealing to diamagnetic property of glass. In this context, the magnetic properties of Ta_2_O_5_ nanocrystal-doped glasses were studied. Figure 11 shows the M versus H loop at 300 K of Ta_2_O_5_ nanocrystal (a) and Ta_2_O_5_-doped glasses (b). It can be seen that the magnetization of BMT in Figure 11a is superparamagnetic without any magnetic saturation and coercivity [4]. Commonly, coercivity *H_c_* is a property of magnetic materials. It is associated with the increase of magnetic anisotropy, which depends not only on intrinsic characteristics of the crystal but also on extrinsic properties, such as shape, size, and doped magnetic ions [42]. In addition, the coercivity increases with magnetic anisotropy because the applied field (at a given temperature) can alter the orientation of magnetization. The energy barrier for coercive can be given by the Equation (8).
(8)$$E_{V}=KVsin^{2}\theta$$
where *K* is the anisotropy constant, *V* is the volume of nanoparticles, and *θ* is the angle between the easy axis and the magnetization direction. It is well known that the magnetic hysteresis behavior of bulk material is strongly influenced by the multi-domain processes like domain–wall displacement and the subsequent realignment of the domain structure [43]. When the material size is reduced to the single-domain size (i.e., <25 nm, in this study), *K* increases remarkably and leads to the increase of the energy barrier *E_v_*, resulting in a sharp decrease of *H_c_* and in the almost zero *H_c_* of superparamagnetic behavior in Figure 11a.

Figure 11b shows the comparison of the magnetization of the glasses. As can be noticed, the magnetization of all the samples is negatively linear to the magnetic field increasing, passing through the zero magnetic field point. This behavior indicates a typical diamagnetic character without any saturation or coercive field because of the inert gas configuration of the host ions. The magnetization increased with the doping amount. The diamagnetic nature of the glass matrix comes from the spin–orbit interaction between the *sp*–*d* band in the diamagnetic Bi^3+^ and Pb^2+^ ions, whose outmost shell has paired electrons. Due to the fully occupied orbitals, such a configuration presents a strong diamagnetic nature. The doping of diamagnetic Ta^5+^ ions increased the concentration of magnetic ions and increased the dipole moment by the spin–orbit interaction between the *d–d* transition of the Ta ions and the *sp–d* interaction between the Bi^3+^, Pb^2+^_,_ and Ta^5+^ ions. The magnetization of the atom mainly came from contributions of the orbitals, electrons, and spin angular momentum. Therefore, the doping of diamagnetic Ta^5+^ ions increases the dipoles and magnetic spin movement, leading to an increase in the magnetization of the glasses.

On the other hand, the increase of magnetization is probably also due to the existence of the nanocrystals. These small nanocrystals inside the matrix acted as a highly polarized single magnetic domain, and the surface anisotropy of nanoparticles dominates the magneto-dynamics [23]. Smaller nanoparticles with a higher surface-to-volume ratio exhibit a much larger proportion of non-compensated surface spins and thus, display a higher magnetization value than those of larger nanoparticles.

Diamagnetic susceptibility is a comprehensive description of a magnetic moment for a free atom having atomic angular moment, electron spin, and diamagnetic response. Diamagnetic behavior is the change in the orbital angular momentum induced by an external magnetic field, and therefore, all materials exhibit a diamagnetic response [44]. Diamagnetic susceptibility is a property of all atoms in molecules and is proportional to the number of electrons and to the square of the radius of the orbit of a closed shell. The influence of magnetic Ta^5+^ ions on diamagnetism can be studied by calculating the diamagnetic susceptibility $x_{D}$ according to the Pascal method, using values for the diamagnetic susceptibility of every atom (*xD_i_*) and bond (*λ_i_*) in the molecule:(9)$$x_{D}=\underset{i}{\sum }xD_{i}+\underset{i}{\sum }\lambda_{i}$$

The $x_{D}$ of Pb^2+^ is (−46 × 10^−6^ emu/mol), of Bi^3+^ is (−25 × 10^−6^ emu/mol), of B^3+^ is (−0.2 × 10^−6^ emu/mol), of O^2−^ is (−6 × 10^−6^ emu/mol), and of Ta^5+^ is (−14 × 10^−6^ emu/mol) [45]. The doping of Ta_2_O_5_ at the expense of B^3+^ in the same molar was helpful for the total diamagnetic susceptibility of the glass based on Equation (9). The theoretically calculated diamagnetic susceptibility of TPB_1_, TPB_5_, and TPB_10_ is 2.46, 5.88, and 9.42 × 10^−6^ emu/mol, respectively, which is very close to that of the magnetization obtained from the experiment measurements (2.21, 5.44, and 9.53 × 10^−6^ emu/mol).

## 4. Conclusions

*5d*^0^ Ta^5+^ in Ta_2_O_5_ has a large density, a big refractive index, and a tunable band gap, which are potentially important for luminescence, nonlinearity, and radiation shielding applications. In this study, Ta_2_O_5_ nanocrystals were synthesized using a hydrothermal method at 600, 800, and 1000 °C. XRD and Raman spectra revealed that the Ta_2_O_5_ prepared at 600 and 800 °C is δ-Ta_2_O_5_ with hexagonal phase, while 1000 °C yielded β-Ta_2_O_5_ with an orthorhombic structure. A high temperature speeds up the crystal growth and the formation of Ta_6_O_12_^6+^ clusters.

Transparent Ta_2_O_5_ nanocrystal-doped heavy metal oxide glasses were obtained by the melt-quenching method. SEM and EDX analysis confirmed that Ta_2_O_5_ exists in the glass matrix (20~30 nm). XRD, FT-IR, and Raman spectra revealed that the Ta^5+^ ions entered into the glass as TaO_6_ octahedral units, which strengthened the network connectivity at 1–5% Ta^5+^ content, while at 10% content, TaO_6_ played a modifier role and distorted the network through producing NBOs and converting BO_4_ to BO_3_ units. The 1% and 5% Ta_2_O_5_-doped glasses exhibited excellent mechanical hardnesses of 477 HV and 479 HV, respectively. The magnetization was greatly enhanced due to the *sp–d* and *d–d* interactions between the Bi^3+^, Pb^2+^_,_ and Ta^5+^ ions. Due to the modification of Ta_2_O_5_, the magnetization $x_{D}$ and radiation shielding efficiency were greatly improved to $x_{D}=$ 9.53 × 10^−6^ emu/mol, *RPE*% = 96–98.8%, and *MAC* = 32.012 cm^2^/g, and *MFP* = 5.02 cm, *HVL* = 0.0035–3.322 cm, and *Z_eff_* = 30.5 due to the increase of polarizability and the Ta_2_O_5_-tuned structure.

## Figures and Tables

**Figure 1 molecules-26-04494-f001:**
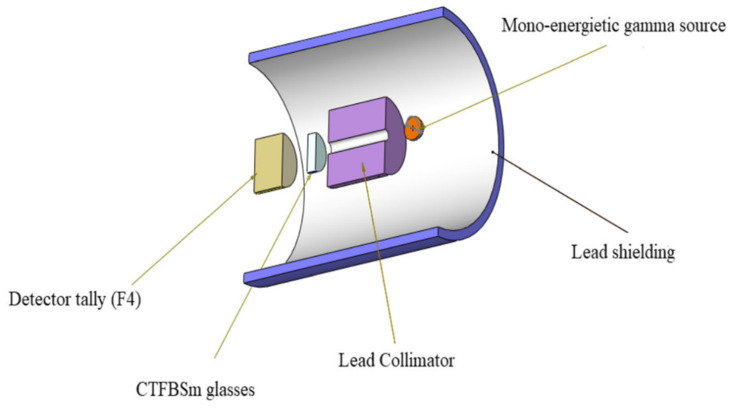
Schematic diagram with simulation geometry of the radiation shielding measurement.

**Figure 2 molecules-26-04494-f002:**
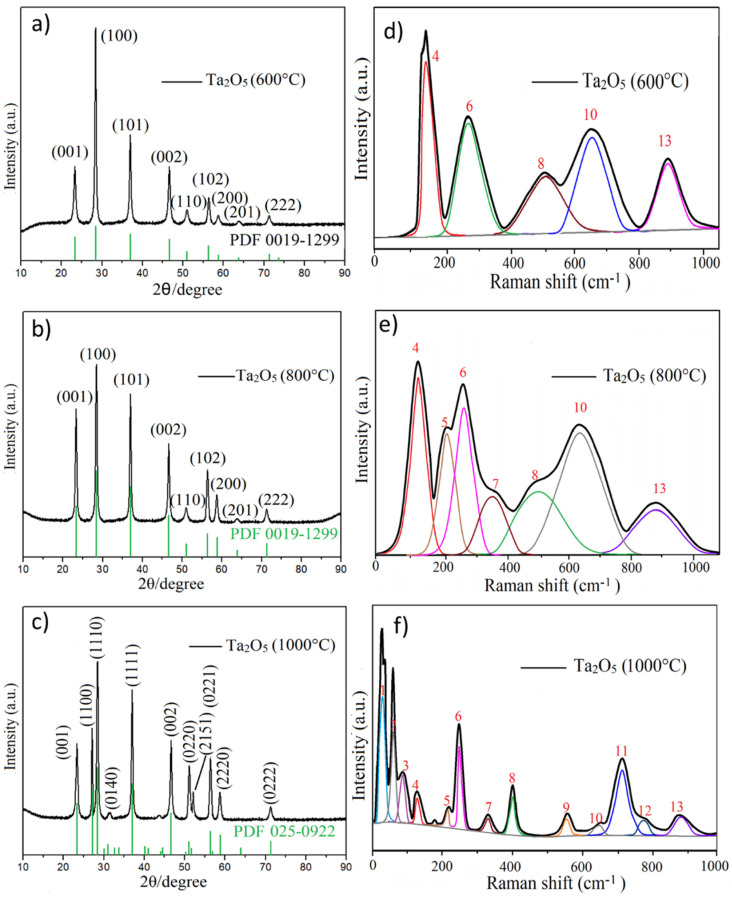
XRD patterns (**a**-**c**) and deconvolution of Raman spectra (**d**-**f**) of Ta_2_O_5_ prepared at 600 °C, 800 °C and 1000 °C, respectively.

**Figure 3 molecules-26-04494-f003:**
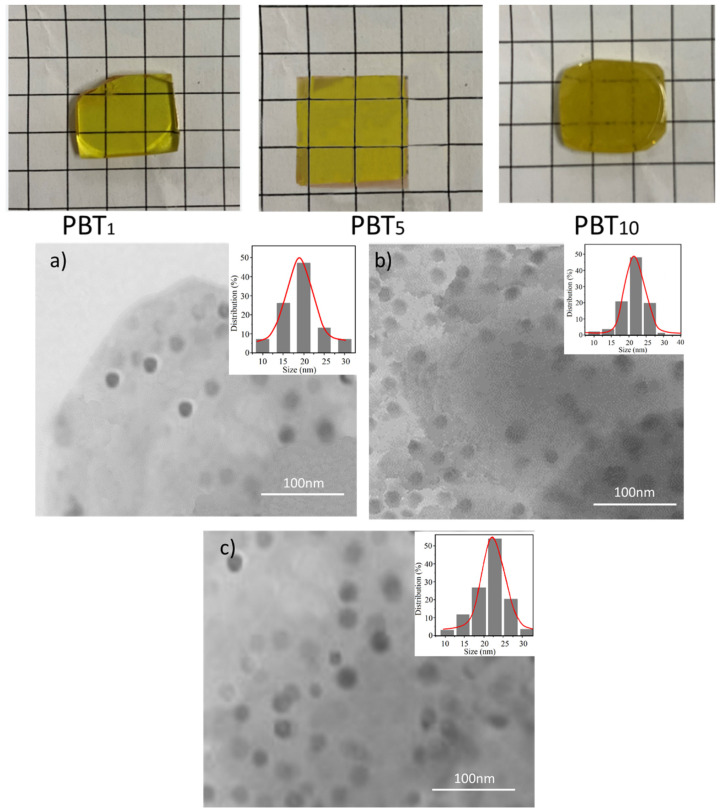
TME images of PBT_1_ (**a**), PBT_5_ (**b**), and PBT_10_ (**c**) and their photographs, the insets are nanocrystal distribution inside glasses.

**Figure 4 molecules-26-04494-f004:**
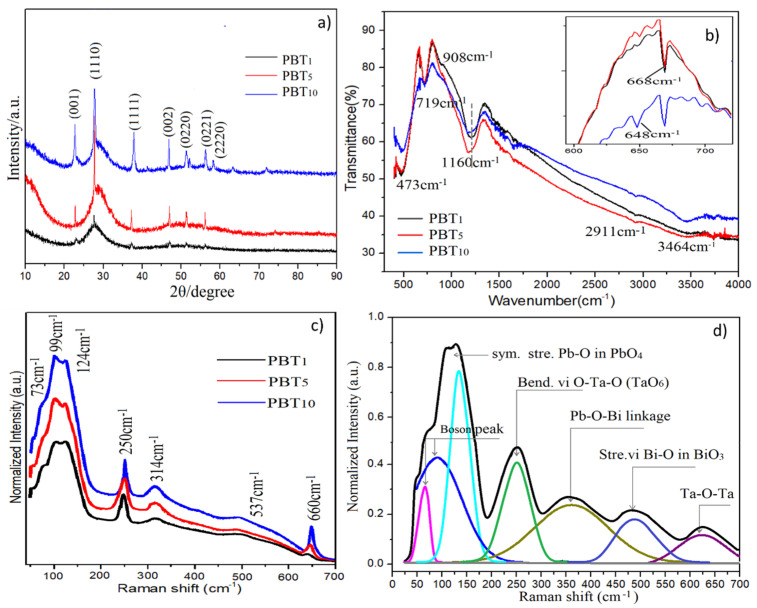
XRD pattern (**a**), FT-IR spectra (**b**), Raman (**c**), and Raman deconvolution (of PBT_10_) (**d**) of three samples.

**Figure 5 molecules-26-04494-f005:**
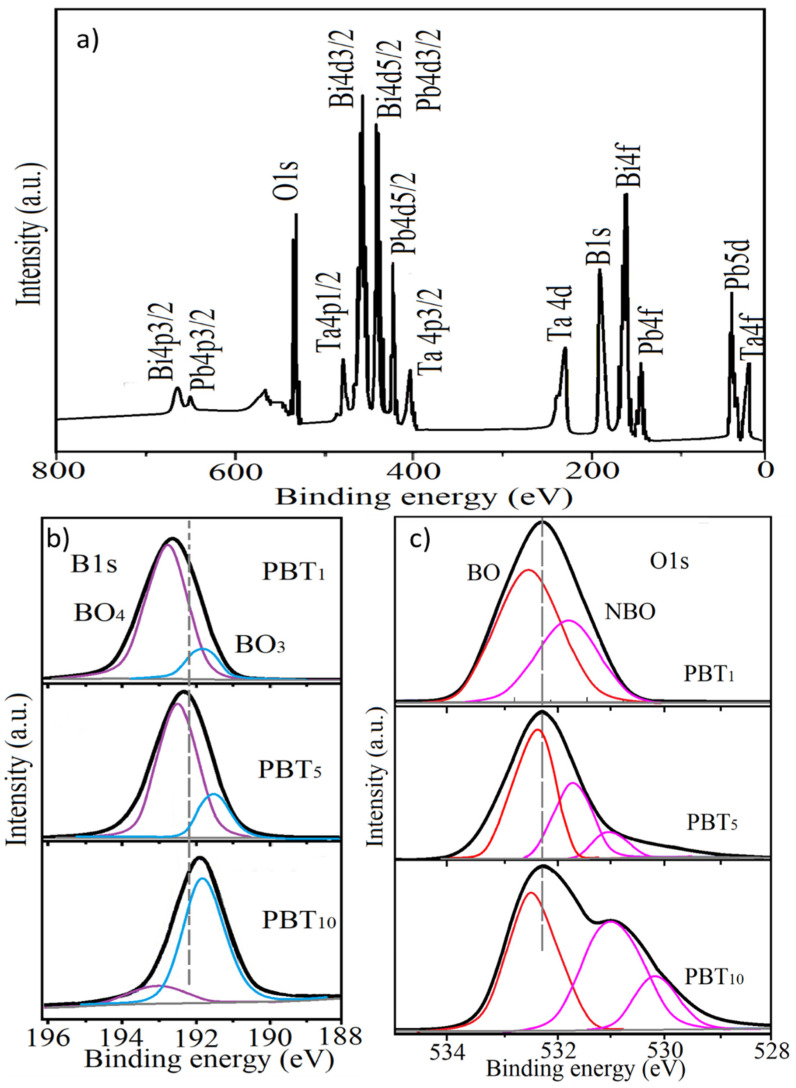
XPS core level energy outline (**a**) of B1s (**b**) and O1s (**c**) of the PBT_1_, PBT_5_, and PBT_10_ glasses, respectively.

**Figure 6 molecules-26-04494-f006:**
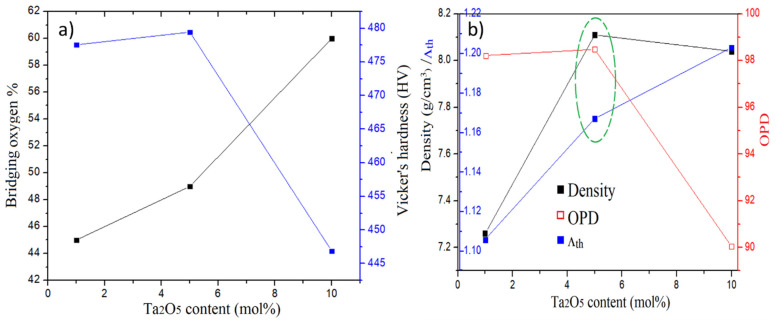
Composition dependence of bridging oxygen numbers, Vickers hardness (**a**) and density, *OPD*, and optical basicity (**b**) of glasses.

**Figure 7 molecules-26-04494-f007:**
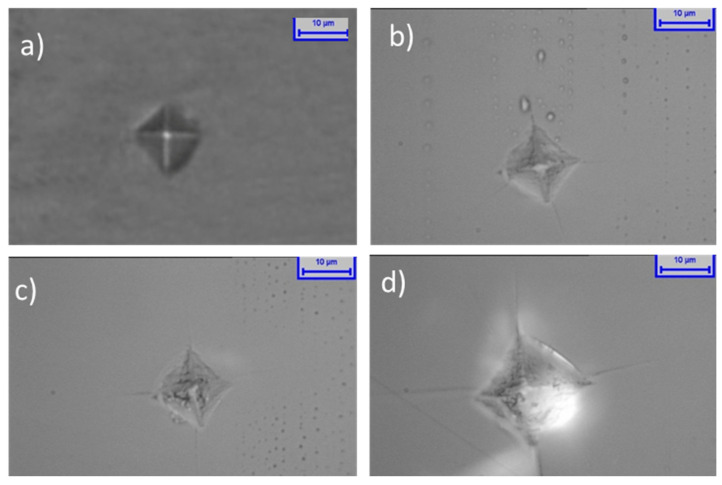
Vickers hardness indentations on the host (**a**), 467.98 HV, PBT_1_ (**b**), PBT_5_ (**c**), and PBT_10_ (**d**).

**Figure 8 molecules-26-04494-f008:**
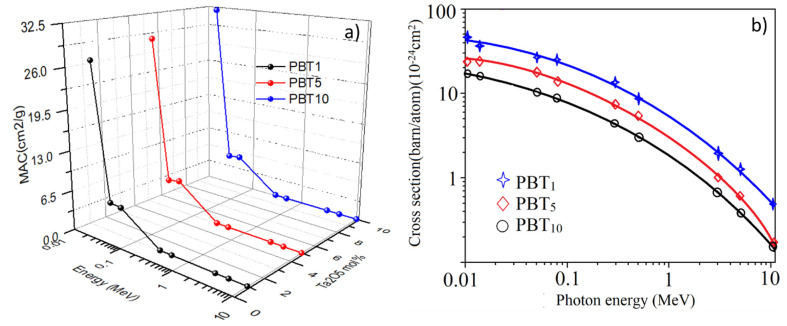
Variation of the *MAC* (**a**) and CS cross-section (**b**) of the fabricated glasses versus the gamma ray energy. Curves in (**b**) are polynomial fitting of the data (blue: y = 41.9813 − 0.016x + 4.23 × 10^−6^ x^2^; red: y = 23.019 − 0.006x + 4.17 × 10^−6^ x^2^; black: y = 12.309 − 0.007x + 5.02 × 10^−6^ x^2^.

**Figure 9 molecules-26-04494-f009:**
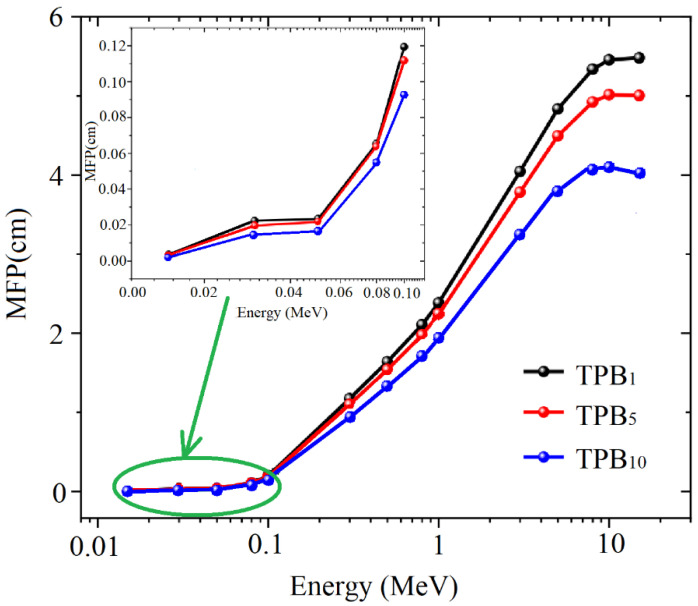
The mean free path (MFP) of glasses at different incident gamma ray energy.

**Figure 10 molecules-26-04494-f010:**
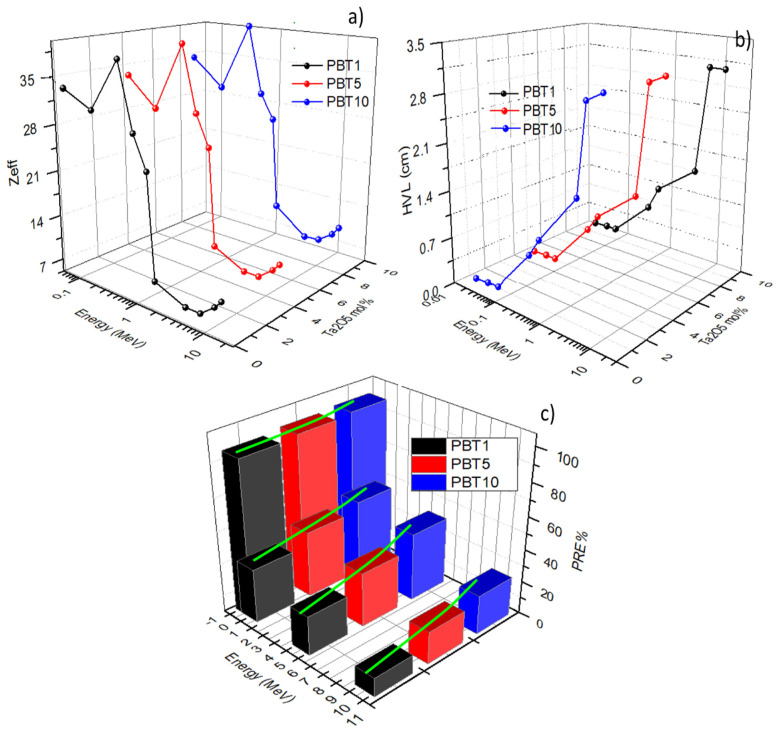
*Z_eff_* (**a**), *HVL* (**b**), and *RPE*% (**c**) of three samples as functions of radiation energy. The green curves in (**c**) are guides.

**Figure 11 molecules-26-04494-f011:**
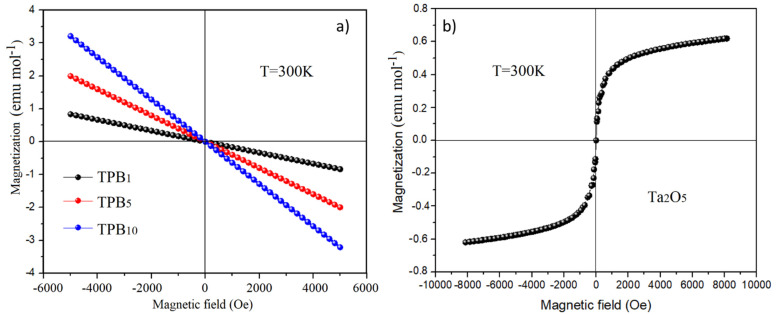
M-H loop of nanocrystals Ta_2_O_5_ (**a**) and Ta_2_O_5_-doped glasses (**b**) at room temperature.

**Table 1 molecules-26-04494-t001:** Raman peak parameters from the deconvolution of the Raman spectrum of the samples.

Bands	Wavenumber (cm^−1^)			Vibration Modes
	600 °C	800 °C	1000 °C	
1	/	/	51	External ionic vibration motion, Ta_6_O_12_^6+^ cluster
2	/	/	78	
3	/	/	94	
4	124	124	124	Bending (deformation) vibration of Ta–O–Ta bonds
5	/	196	245	
6	269	269	269	
7	/	/	338	
8	499	494	408	Stretching vibration of Ta–O bonds
9	/	/	562	
10	642	642	642	
11	/	/	711	
12	/	/	780	
13	883	883	883	Stretching vibration of Ta–O–Ta bonds

**Table 2 molecules-26-04494-t002:** The ratio of BO_4_/BO_3_ and BO/NBO from the calculation of the XPS deconvolution area, Vickers hardness of samples (HV), density (ρ), oxygen packing density (*OPD*), and optical basicity (ᴧ_th_).

Samples	BO_4_/BO_3_	BO/NBO	ρ (gcm^−3^) ± 0.01	*OPD* ± 0.01	HV ± 0.01	ᴧ_th_ ± 0.001
PBT_1_	78/22	55/45	7.26	98.22	477.55	1.107
PBT_5_	76/24	53/47	8.11	98.48	479.44	1.168
PBT_10_	15/85	40/60	8.14	90.05	446.88	1.204

**Table 3 molecules-26-04494-t003:** Mass attenuation coefficients (*MAC*) (${\mathrm{cm}}^{2}/g$) comparison of the MCNP and XCOM of the glasses.

Energy (MeV)	PBT_1_			PBT_5_			PBT_10_		
	MCNP	XCOM	RD (%)	MCNP	XCOM	RD (%)	MCNP	XCOM	RD (%)
0.015	26.854	27.002	0.022	28.633	28.712	0.016	32.003	32.012	0.014
0.05	4.841	4.874	0.832	4.944	4.963	0.842	5.211	5.241	0.811
0.08	4.638	4.750	0.797	5.309	5.505	0.718	5.4422	5.6533	0.38
0.3	0.122	0.132	0.819	0.123	0.135	0.975	0.146	0.156	0.224
0.5	0.088	0.089	0.011	0.090	0.091	0.011	0.090	0.092	0.022
3	0.050	0.051	0.291	0.060	0.062	0.272	0.061	0.063	0.413
5	0.027	0.027	0.228	0.028	0.028	0.231	0.028	0.030	0.230
10	0.026	0.026	0.191	0.027	0.027	0.183	0.028	0.029	0.172

## Data Availability

Not applicable.

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
