# Peer review of "Ta2O5 Nanocrystals Strengthened Mechanical, Magnetic, and Radiation Shielding Properties of Heavy Metal Oxide Glass"

_molecules, 2021, doi:10.3390/molecules26154494_

Round 1

Reviewer 1 Report

This manuscript entitled "Ta2O5 nanocrystals strengthened mechanical, magnetic and radiation shielding properties of heavy metal oxide glass" has been reviewed, though, the authors carefully describe the crystal and physical properties of inorganic materials. However, the writing that lacks logical structure and excessive typos makes it difficult to associate its scientific value.

In abstract, what is "orthorhomTa2O5"? is "orthorhombic Ta2O5". What is "heavy metal oxide" in glass"es"?

These terms should be well-defined.

In line 49, being an ‘modifier’ oxide?

In Fig.4, what means the detector totally(F4), please explain it?

In table 3, the unit "μm (〖cm〗^2/g)" is confused.
Authors should focus on the potential applications of this material, more logically, rather than piling up the measurement data. Personally suggest that this article can be published in journals related to applied materials after revision.

Author Response

Response to reviewer’s comments

Comments and Suggestions for Authors

This manuscript entitled "Ta2O5 nanocrystals strengthened mechanical, magnetic and radiation shielding properties of heavy metal oxide glass" has been reviewed, though, the authors carefully describe the crystal and physical properties of inorganic materials. However, the writing that lacks logical structure and excessive typos makes it difficult to associate its scientific value.

Point 1:

In abstract, what is "orthorhomTa2O5"? is "orthorhombic Ta2O5". What is "heavy metal oxide" in glass"es"?

These terms should be well-defined.

Response:  sorry for this mistake. Yes, it should be orthorhombic Ta2O5. The author would like to explain that the glasses are heavy metal oxide glasses.  When the heavy metal oxides such as PbO and Bi2O3 are the main components of glass, we call these glass are heavy metal oxide glasses or heavy metal oxide based glasses. This kind of glass contains more than 50 % in total of heavy metal oxides and at least 10% of glass former such as B2O3 and SiO2, because the heavy metal oxides are not traditional glass former, they cannot form glass by themselves. Therefore, they need traditional glass former to form the glass network. The author thanks a lot the comments.

Point 2:

In line 49, being an ‘modifier’ oxide?

Response: yes, compared with the conventional glass former, the Ta2O5 is a conditional glass former, and in many cases it acts as the modifier role in glass forming. That is why we call it “modifier” oxide.

Point 3:

In Fig.4, what means the detector totally(F4), please explain it?

Response: yes, in the text, the description is like this: “In the present simulation process, the detector was considered F4 tally to estimate the incident gamma photons’ average track length. The detector was set up to be at a distance of 15 cm of the mono-energetic source. “

The system includes a radiation source, glass in this study acts as shieling material, Pb blocks to avoid the scattered photons, the glass is located between the radiation source and F4 tally mesh detector, and the distance between detector and source is 15cm in this study. Therefore, the F4 tally mesh is used as detector to count the incident high-energy photons intensities and estimate the average trace length.  

In the revised version, authors would like to modify this sentence to make it more clear to readers.

“In the present simulation process, glass in this study acts as shieling material while bulk Pb blocks to avoid the scattered photons. The F4 tally mesh detector was used to count the high-energy photons intensities and estimate the average track length. The glass is located between radiation source and detector, and the distance between source and detector is 15 cm.”     

For further information about the tally F4:

Tally F4 is the averaged track length in a predefined volume divided by volume for one primary particle but we can use it freely as volume flux.  F4 tally is the average flux in a cell. we can obtain the average flux in output data. Here we can see the F4 tally appropriate syntax:

Mode p
mode p sdef pos=0 0 100 axs=0 0 -1 erg=0.120 par=2  
f4:p 4

m1 1000 .040402 6000 .123247 7000 .011862 &
8000 .357394 74000 .467094
ctme 1

Therefore, if we give the command the input that "Consider the Photon and Give me the average flux in cell 4" and also when we look at the output file, we will see an average value in the bottom of the cell part. This is our average flux. that is a preset code in the MCNP simulation system.

Point 4:

In table 3, the unit "μm (〖cm〗^2/g)" is confused.

Response: thanks for this question. The μm refers to the Mass attenuation coefficient (MAC), not μm (micron). The unit of Mass attenuation coefficient is .

Yes, this is a little confused to reader, in the revised version, author would like to use the MAC to replace the μm. many thanks.

Point 5:

Authors should focus on the potential applications of this material, more logically, rather than piling up the measurement data. Personally suggest that this article can be published in journals related to applied materials after revision.

Response: yes, thanks for this suggestion. Authors made many corrections to strengthen the logic of this article, and some of these modification are listed below:

  • The introduction section has been re-organized to make the whole article more logic.
  • The physical property of nanocrystals doped glasses depends on the size, shape and property of nanocrystals. Therefore, firstly we investigate the structure of synthesized nanocrystal. Figs.2a-2c show the XRD pattern of samples prepared…..
  • Raman spectroscopy is commonly used in chemistry to provide a structural fingerprint by which molecules can be identified. In order to give in-depth study on the structure of Ta2O5 nanocrystals, Raman spectra were recorded which provided characteristic vibration modes of Ta2O5 Figs.2d -2f show the Raman spectra of Ta2O5 prepared…..
  • Therefore, from the comparison of XRD and Raman spectra of three samples, the Ta2O5 prepared at 800°C exhibits promising size, morphology and structure and this sample is used as dopant to glasses.
  • The morphology and nanocrystals distribution inside glass is important for glass properties. Therefore, TEM images of glasses were taken to provide the morphology and distribution of nanocrystals. As mentioned in the experimental part…….
  • The structure of glass after doping of Ta2O5 nanocrystals inside glasses is influential to properties. The structures of glasses were studied using XRD, FT-IR and Raman spectra….
  • From the above studies on density, OPD and hardness, it is found that that Ta2O5 nanocrystals enhanced the compactness and mechanical performance which are helpful to improve the radiation shielding as well. Therefore, the radiation shielding properties were studied among which the attenuation coefficient is important. The linear attenuation coefficient values (µ)……
  • Usually materials doped with large mass dopant such as Ta2O5 exhibit an improved polarization character which is helpful to magnetic and magneto optical performance. On the other hand, glass having good magnetic and large compactness property has aroused big interests in recent year for radiation shielding and magneto-optical devices. In case of this study, Ta5+ with an empty d shell (d0 ions) having completely occupied outermost electronic shell is attractive to increase the diamagnetic property of glass. In this context, the magnetic properties of Ta2O5 nanocrystals doped glasses have been studied. Fig. 11 shows the M versus H loop at 300K of nanocrystal…..
  • Due to the modification of Ta2O5, the magnetizationand radiation shielding efficiency were greatly improved to 53 ×10-6 emu/mol,  RPE%=96-98.8% and MAC=32.012 cm2/g, MFP= 5.02cm, HVL=0.0035-3.322cm and Zeff =30.5 due to the increase of polarizability and Ta2O5 tuned structure.

Reviewer 2 Report

The manuscript is clear, well arranged, the experimental results are clearly presented and discussed.

Author Response

Response to reviewer’s comments

Comments and Suggestions for Authors

Point 1:

The manuscript is clear, well arranged, the experimental results are clearly presented and discussed.

Response: the authors feel very appreciated to the positive evaluation on the article, and many thanks for this encouragement from respected reviewer to authors.

Reviewer 3 Report

The paper titled "Ta2O5 nanocrystals strengthened mechanical, magnetic and radiation shielding properties of heavy metal oxide glass" by X. Zhang et al. provides a good and comprehensive analysis of diamagnetic 5d0 Ta5+ ions and Ta2O5 nanocrystals used to enhance several important characteristics of heavy metal oxide glasses. The structure of doped glasses was properly assessed through X-ray diffraction, Infrared spectroscopy and Raman spectroscopy, evidencing the presence of TaO6 octahedral units in the Ta2O5 nanocrystals which act as network modifier. Furthermore, the high density and polarizability of Ta2O5 nanocrystals provided enhanced Vicker’s hardness, magnetization and radiation shielding behaviors, furnishing an experimental proof of the effectiveness of such materials for different applications.

In my opinion the paper is well-written, of high technical quality, and includes all appropriate references for further details. The reported data are accurate and exhaustively explained, providing a comprehensive and detailed scenario about the specific correlations between the composition/structure of the investigated samples and the mechanical/magnetization/radiation shielding efficiency enhancement observed.

Based on the aforementioned considerations, the paper molecules-1292161 certainly deserves, in my opinion, publication in the journal Molecules.  

Few minor comments to further improve this paper are listed below: 

  1. Please, provide uniform unit format in Figure 1: Intensity/a.u. or Intensity (a.u.).
  2. At lines 202-205 please specify better the difference in the Raman profiles observed between the Ta2O5 prepared at 1000°C, 800 and 600 °C, as the “bands 1-3 assigned to lattice modes are active in orthorhombic Ta2O5 prepared at 1000°C, while on the contrary, they are inactive for sample prepared at 600 and 800°C, indicating that Ta polyhedral and Ta6O126+ clusters exist only orthorhombic Ta2O5” statement appears quite confusing and not fully exhaustive.
  3. Please specify the full name of PBT, as it appears, for the first time, at line 218.
  4. Lines 246-247: “Another two peaks at 2911 and 3463 cm-1 are related to the bending and stretching vibration of OH bonds in samples, respectively”. Please, include further details about the provenance of such hydroxyl groups within the investigated samples. Their possible provenance/origin in Ta2O3 doped glasses is not clear to me. Furthermore, on the basis of which criteria the authors assigned the band at ~2911 cm-1 to the bending vibration of OH bonds? It is well-known that it commonly falls at ~1630 cm-1. Please provide also a comment on this.
  5. Line 465: authors refer to Fig.9e, however, neither the e panel nor the green curve are present. Please check.
  6. There are lot of formatting issues (empty spaces, unnecessary underlines, etc.). In addition, in the author’s affiliations (page 1), remove the “2” from “2Material Sciences & Engineering department, Henan University of Technology, Lianhua road 100, 450001, 6 Zhengzhou, China”. Finally, in the reference list, the numbering is repeated twice.
  7. Although the writing looks pretty good to me, often an incorrect choice of wording is chosen. Rewriting with a native English speaker is recommended.

Author Response

Response to reviewer’s comment

Comments and Suggestions for Authors

The paper titled "Ta2O5 nanocrystals strengthened mechanical, magnetic and radiation shielding properties of heavy metal oxide glass" by X. Zhang et al. provides a good and comprehensive analysis of diamagnetic 5d0 Ta5+ ions and Ta2O5 nanocrystals used to enhance several important characteristics of heavy metal oxide glasses. The structure of doped glasses was properly assessed through X-ray diffraction, Infrared spectroscopy and Raman spectroscopy, evidencing the presence of TaO6 octahedral units in the Ta2O5 nanocrystals which act as network modifier. Furthermore, the high density and polarizability of Ta2O5 nanocrystals provided enhanced Vicker’s hardness, magnetization and radiation shielding behaviors, furnishing an experimental proof of the effectiveness of such materials for different applications.

In my opinion the paper is well-written, of high technical quality, and includes all appropriate references for further details. The reported data are accurate and exhaustively explained, providing a comprehensive and detailed scenario about the specific correlations between the composition/structure of the investigated samples and the mechanical/magnetization/radiation shielding efficiency enhancement observed.

Based on the aforementioned considerations, the paper molecules-1292161 certainly deserves, in my opinion, publication in the journal Molecules.  

Response: the authors feel very appreciated to the positive evaluation on the article, and many thanks for this encouragement from respected reviewer to authors.

Few minor comments to further improve this paper are listed below: 

Point 1:

Please, provide uniform unit format in Figure 1: Intensity/a.u. or Intensity (a.u.).

Response: yes, we should use the uniform unit format in the whole manuscript. We have revised these mistakes. Many thanks!!

Point 2:

At lines 202-205 please specify better the difference in the Raman profiles observed between the Ta2O5 prepared at 1000°C, 800 and 600 °C, as the “bands 1-3 assigned to lattice modes are active in orthorhombic Ta2O5 prepared at 1000°C, while on the contrary, they are inactive for sample prepared at 600 and 800°C, indicating that Ta polyhedral and Ta6O126+ clusters exist only orthorhombic Ta2O5” statement appears quite confusing and not fully exhaustive.

Response: sorry for this confusing statement on the band 1 to band 3. Yes, we would like to address this sentence like this “It can be seen from Fig.3 that the lowest wavenumber vibration bands 1 to band 3 are sharp and intense in Ta2O5 prepared at 1000°C, while these bands disappear for Ta2O5 prepared at 600°C and 800°C. This result indicates that the Ta polyhedral and Ta6O126+ clusters existed in Ta2O5 prepared at 1000°C due to the higher temperature.” 

Point 3:

Please specify the full name of PBT, as it appears, for the first time, at line 218.

Response: sorry for this careless mistake. Yes, the PBT glass should be defined in advance and then can be used in the text. Sorry for this. In the revised version, authors have addressed the full name of PBT glass in the experimental section. “The stoichiometric compositions of the batch materials (20 g) 40PbO-45Bi2O3-(15-x) H3BO3- xTa2O5, where x = 1, 5 and 10 mol%, corresponding glasses are labeled as PBT0 (host), PBT1, PBT5 and PBT10.”

Point 4:

Lines 246-247: “Another two peaks at 2911 and 3463 cm-1 are related to the bending and stretching vibration of OH bonds in samples, respectively”. Please, include further details about the provenance of such hydroxyl groups within the investigated samples. Their possible provenance/origin in Ta2O5 doped glasses is not clear to me. Furthermore, on the basis of which criteria the authors assigned the band at ~2911 cm-1 to the bending vibration of OH bonds? It is well-known that it commonly falls at ~1630 cm-1. Please provide also a comment on this.

Response:  firstly, authors feel sorry for the mistakes statement on the OH groups bending vibrations. Usually the OH stretching vibration appears from 2900-3500cm-1 therefore, in this study, the very weak peak at 2911 and 3463cm-1 are associated with the stretching vibration of OH groups of water in glasses. Compared with the OH stretching vibration, only a limited number of studies have been devoted to the OH bending vibration. This is because the OH bending vibration has the lowest frequency among the intramolecular modes, and therefore, energy relaxation of this mode should involve other intermolecular vibrations which makes it more complicated to depict. Yes, usually the OH bending vibrations sometimes appears around 1625-1635cm-1 [a-c], but sometimes appears at 1645cm-1 [d] and 1650cm-1 [e].

The presence of OH bonds in Ta2O5 doped glasses mainly because the glasses are fabricated in air, not in glove box with vacuum. Most glasses fabricated in air contain the OH groups which is not helpful for photonics fiber fabrication. In that case, the glass should be fabricated in a vacuum glove box full with N2 gas.  

Therefore, in the revised version, authors would like to address the sentence like this:” Considering that all glasses were fabricated in air, water can be presented in glasses. Another peaks at 2911 (very weak) and 3463 cm-1 are related to the stretching vibration of OH bonds in samples. A weak peak around 1630cm-1 is due to the bending vibration mode of OH groups [a].”

[a] Kim S. Finnie, David J. Cassidy, John R. Bartlett, and James L. Woolfrey, IR Spectroscopy of Surface Water and Hydroxyl Species on Nanocrystalline TiO2 Films, Langmuir 2001, 17, 816-820

[b] Roger D. Aines and George R. Rossman, Water in mineral? A peak in the infrared, Journal of Geophysical research, 89 (1984) 4059-4071 

[c] M Mirzan, K Wijaya, I I Falah, W Trisunaryanti, Synthesis and characterization of Ni-promoted zirconia pillared bentonite, Journal of Physics: Conf. Series 1242 (2019) 012013

[d] Ashihara, S.; Fujioka, S.; Shibuya, K. Temperature Dependence of Vibrational Relaxation of the OH Bending Excitation in Liquid H2O. Chem. Phys. Lett. 2011, 502, 57–62.

[e] Takakazu Seki, Kuo-Yang Chiang, Chun-Chieh Yu, The Bending Mode of Water: A Powerful Probe for Hydrogen Bond Structure of Aqueous Systems, J. Phys. Chem. Lett., 11, 19(2020) 8459–8469

Point 5:

Line 465: authors refer to Fig.9e, however, neither the panel nor the green curve are present. Please check.

Response: sorry for this typewriting mistake. It should be Fig.10c, not Fig.9e. in the revised version, this error has been corrected and highlighted with red color. Thanks a lot!!

Point 6:

There are lot of formatting issues (empty spaces, unnecessary underlines, etc.). In addition, in the author’s affiliations (page 1), remove the “2” from “2Material Sciences & Engineering department, Henan University of Technology, Lianhua road 100, 450001, 6 Zhengzhou, China”. Finally, in the reference list, the numbering is repeated twice.

Response: yes, all these errors and other formatting issues have been corrected in the revised version with red color. thanks a lot!!

In the first page, the affiliations and reference numbering are found to be normal. Probably due to the problem occurring during pdf formation. Anyway, in the revised version, these are correct in format.

As for the red underlines of some words such as coercivity and so on, these probably due to the version mismatch however, the words are correct. Authors sincerely beg kindly understanding. Thanks a lot!

Point 7:

Although the writing looks pretty good to me, often an incorrect choice of wording is chosen. Rewriting with a native English speaker is recommended.

Response: many thanks for this good advice.  The revised version has been checked by one native English speaker friend. Many thanks for all of your comments and your encouraging evaluation.

Round 2

Reviewer 1 Report

Thanks to the authors for their efforts to improve the quality of the draft. However, the Table 3 and Figure 10 (a), (b), (c) are visualized and wordy presentations. In my personal opinion, the authors’ writing has piled up too much unnecessary information, and the publication standards of "Molecules" have not yet been reached.